# Influence of Nociception Level Monitor (NOL)-Guided Analgesic Delivery on Perioperative Course in Breast Surgeries: A Randomized Controlled Trial

**DOI:** 10.3390/medicina60121921

**Published:** 2024-11-22

**Authors:** Laima Malachauskiene, Rajesh Bhavsar, Skule Bakke, Jeppe Keller, Swati Bhavsar, Anne-Marie Luy, Thomas Strøm

**Affiliations:** 1Department of Anesthesia and Critical Care Medicine, South Jutland Hospitals, South Denmark University, Kresten Philipsens Vej 15, 6200 Aabenraa, Denmark; laima.malachauskiene@regionh.dk (L.M.); skule.arnesen.bakke1@rsyd.dk (S.B.); jeppe.keller@rsyd.dk (J.K.); swati.rajesh.bhavsar@rsyd.dk (S.B.); annemarie.luy@rsyd.dk (A.-M.L.); 2Department of Anaesthesia and Intensive care, Odense university hospital, 5000 Odense, Denmark; thomas.stroem@rsyd.dk

**Keywords:** objective nociception monitoring, opioid consumption, postoperative nausea vomiting, postmastectomy pain syndrome

## Abstract

*Background and Objectives:* Breast cancer surgeries offer challenges in perioperative pain management, especially in the presence of inherent risk of postoperative nausea and vomiting (PONV) and postmastectomy pain syndrome (PMPS). Inappropriate opioid consumption was speculated as one of the reasons. Through this study, the influence of objective pain monitoring through a nociception level monitor (NOL) on perioperative course in breast surgeries was investigated. *Materials and Methods:* This was a prospective randomized study conducted at a regional hospital. Sixty female patients posted for breast cancer surgery were randomized equally into study and control groups. Both groups were monitored using BIS and NOL, but in the control group, the NOL monitor was blinded by a cover. Both groups received propofol and remifentanil through target-controlled infusions (TCIs) along with interpectoral, pectoserratus (PECS II), and superficial pectointercostal block. The primary outcome was intraoperative opioid consumption. Secondary outcomes were PONV, eligibility for discharge from the recovery room, and symptoms of PMPS after three months. *Results:* Two patients were excluded. The study group received significantly less remifentanil (0.9 mg in the study group vs. 1.35 mg in the control group, *p* = 0.033) and morphine (2.5 mg in study group vs. 5 mg in control group, *p* = 0.013). There was no difference in PMPS symptoms between the groups. The study group showed longer duration of inadequate analgesia (i.e., 7% vs. 10% of the total intraoperative period in control and study group, respectively, *p* = 0.008). There was no difference in time to eligibility for discharge from the recovery room between the groups. *Conclusions:* NOL monitor-guided analgesic delivery reduces intraoperative opioid consumption. No difference was demonstrated on PONV, eligibility for discharge from the recovery room, or PMPS symptoms.

## 1. Introduction

For decades, intraoperative analgesia has been administered empirically, often relying on nonobjective surrogate parameters such as blood pressure and heart rate. This approach may lead to hemodynamic instability, poor recovery, and patient dissatisfaction [1,2,3]. In response, objective pain monitoring methods were sought for a long time. A recently developed nociception monitor (NOL) has shown promising results in various intraoperative situations [4,5,6,7].

Breast cancer patients face unique perioperative pain management challenges, as they have multiple inherent risk factors for postoperative nausea and vomiting (PONV) and postmastectomy pain syndrome (PMPS) [8,9,10]. PMPS often includes symptoms like paresthesia, numbness, mild to moderate pain around the incision site, and difficulty in lifting the arm(s), and can persist for more than three months after surgery [11]. Apart from surgery-related complications such as nerve and muscle injury, scar formation, and prolonged inflammatory response, inappropriate opioid administration has been suggested as one of the possible contributing factors [12]. Various strategies, including ultrasound-guided interpectoral, pectoserratus (PECS II), and superficial pectointercostal (parasternal) blocks, have been explored to minimize opioid use [13,14]. Due to their shortcomings, peripheral nerve blocks are yet to be established as an exclusive anesthesia technique for breast surgeries [15]. Therefore, major breast surgical procedures are routinely performed under general anesthesia with or without supplementation of regional blocks. It is important to observe that opioids are still administered during anesthesia due to the absence of objective pain assessment and fear of awareness.

With the availability of novel NOL monitoring, clinicians are able to adjust opioid administration more precisely, potentially reducing the risks associated with empirical dosing. In this randomized controlled study, we aimed to evaluate the effect of NOL-guided analgesia on intraoperative opioid consumption and its influence on long-term outcomes like PONV and PMPS.

## 2. Materials and Methods

### 2.1. Study Design

This was a prospective randomized controlled study conducted at a single regional hospital. Patients were randomized using Redcap data management software (last updated on march 2022) to either an NOL monitor-guided analgesia group (study group) or the standard-care group (control group). The trial design and reporting followed the Consolidated Standards of Reporting Trials (CONSORT) guidelines to ensure rigorous reporting of randomized controlled trials [16].

### 2.2. Ethics

The study was conducted according to the guidelines of the Declaration of Helsinki and approved by the Medical Research Ethics Committee of South Denmark. (project ID: S-20210083/PBH/csf, approved on 2 November 2021). The study was registered at a local university and on a public registry. There were no amendments to the study protocol. Informed consent was obtained from all subjects involved in the study. All patients were enrolled after receipt of permission to follow-up by telephone for symptoms of PMPS at the end of 3 months.

### 2.3. Participants

Female patients (aged 18 to 80 years) with ASA classification I–IV scheduled for elective mastectomy for breast cancer were recruited. Exclusion criteria included inability to consent, atrial fibrillation, local anesthetic allergy, and procedures converted from lumpectomy to mastectomy.

### 2.4. Interventions

All patients received a usual anesthesia regimen, which included target-controlled infusion (TCI) of propofol and remifentanil together with interpectoral and pectoserratus (PECSII) accompanied by superficial pectointercostal blocks. In the study group, the NOL monitor controlled the remifentanil TCI rates, and the target NOL index was kept in the range of 10–27. The control group received analgesia based on the judgment of the treating anesthesiologist with no NOL guidance, as the monitor data were blinded to the anesthetist.

### 2.5. Outcome Measures

The primary outcome was intraoperative opioid consumption, specifically remifentanil and morphine usage. Secondary outcomes included time to eligibility for discharge from the RR, RR stay duration, length of hospital stay, incidence of PONV, and the presence of PMPS symptoms at three months post-surgery.

### 2.6. NOL Monitoring and Analgesia Protocol

The NOL monitor (PMD-200, Medasense Biometrics Ltd., Ramat Gan, Israel) provided real-time assessments of nociceptive responses using parameters like photoplethysmogram amplitude, skin conductance, and heart rate variability to calculate a numerical value, i.e., the NOL index. This index ranges from 0 to 100, where lower values (less than 10) represent the absence or lower grades of noxious stimulation and may be understood as excessive analgesia. Values of 10–27 may suggest an adequate nociception–antinociception balance state, while values greater than 27 signify the presence of severe noxious stimulation. The NOL monitor senses the relevant parameters continuously to calculate and register the NOL index every 5 seconds. At the end of the procedure, the monitor analyzes all values of the NOL index collected throughout the procedure and categorizes them as per the analgesia levels, i.e., <10 (excessive analgesia), 10–27 (adequate analgesia), and >27 (inadequate analgesia), with their duration as a percentage of the entire procedure.

In the study group, adjustments to remifentanil were made only if the NOL index deviated from the target range for over two minutes.

### 2.7. RRD Score

The RRD score was used as a tool to define the accurate and objective time at which the patient was eligible for discharge from the RR. The RRD reflects the objective clinical status of the patients, as local logistics and the judgment of the treating physician may influence the actual discharge of the patients from the RR. The RRD is a scoring system developed for general surgery by the Danish Society of Anesthesia and Intensive Care [17] (Appendix A).

### 2.8. Standard Anesthesia Regimen for Both the Groups

All patients received 1 g paracetamol orally on the day of the surgery. On the operation day, after insertion of an intravenous canula, standard monitors, such as a 3-lead continuous electrocardiogram, noninvasive blood pressure, pulse oximetry, and BIS monitoring were connected. Additionally, all patients were connected to an NOL monitor. The sensor was positioned on the middle finger of the hand contralateral to the blood pressure cuff. In the study group, the NOL monitor was used to steer analgesia during surgery, while in the control group, the monitor was covered and not used for clinical decision-making.

In all patients, anesthetic induction was carried out using TCI of propofol and remifentanil. For the delivery of anesthetic drugs, two separate infusion pumps (BD Alaris^™^ neXus CC Syringe Pump), one programmed with the remifentanil pharmacokinetic dataset of Minto et al. and another programmed with the propofol pharmacokinetic dataset of Marsh et al., were used [18,19]. For induction, the target effect site concentration for remifentanil was set at 8 ng/mL, and for propofol, it was set at 8 μg/mL. After loss of consciousness (as detected by BIS values below 50, absence of the eyelash reflex, and no response to verbal stimulation), the airway was secured with a supraglottic airway device, i.e., a laryngeal mask airway (LMA). The ventilator settings were programmed such that the end-tidal PCO2 was kept at 4.5 ± 0.4 vol.% (34 ± 3 mm Hg). After insertion of the LMA, the target propofol concentration was adjusted in steps of 0.5 μg/mL to ensure a steady-state BIS value of 50 ± 5. The depth of anesthesia was maintained (BIS 50 ± 5) by adjusting the rate of propofol infusion. All patients received dexamethasone (4 mg) intravenously.

### 2.9. Block Performance

As a supplement to general anesthesia, interpectoral and pectoserratus blocks were administered after induction of anesthesia and placement of the LMA. Ropivacaine (0.2%, 20 mL) was injected in each plane using an 80 mm-long peripheral nerve block needle (SonoTAP II, Pajunk) under the guidance of a linear ultrasound probe (L5-14 MHz, SECMA, Sonosite) using the in-plane technique. The local anesthetic was expected to spread between the clavipectoral fascia and the superficial border of the serratus anterior muscles. The medicine was expected to anesthetize the anterior cutaneous branches of intercostal nerves III–IV–V–VI situated between the thoracic spinal nerve (T4–T6), the intercostobrachial, and the long thoracic nerve [20,21]. The block was expected to offer analgesia over the central and lateral part of the surgical area during the breast surgery [15]. It was accompanied by superficial pectointercostal block, where 10 mL of 0.2% ropivacaine was injected under the pectoralis major muscle just lateral to the sternum at the third intercostal space in both cephalad and caudal directions (5 mL each) using a separate block needle (50 mm, SonoTAP II, Pajunk). The block was expected to provide analgesia on the medial part of the incision site.

The surgical incision was made 15 min after administration of the block.

### 2.10. Control Group

In control group patients, the TCI remifentanil rate was adjusted as per the judgment of the attending anesthetist. At the end of the surgery, morphine (10 mg) was injected for postoperative pain as a preemptive analgesic, as the peripheral nerve blocks may not anaesthetize all the nerves in the surgical area. Despite blocks, surgical work deep in the axillary region may cause severe pain, which may be masked during surgery by remifentanil, but may surface in the immediate postoperative period, especially when the patient is awake [22]. Treatment of hemodynamic changes such as bradycardia, tachycardia, increased blood pressure, or hypotension were at the discretion of the attending anesthesiologist. This included an increase or decrease in the TCI rate of propofol or remifentanil, fluid supplementation, or IV administration of ephedrine or atropine wherever suitable.

### 2.11. Study Group

The rate of TCI remifentanil was adjusted under the guidance of the NOL monitor to keep the index between 10 and 25. The infusion rate was adjusted only when the change in the NOL persisted for more than 2 min. At the completion of surgery, the TCI remifentanil was stopped, and after 10 min, in contrast to the control group, IV morphine was administered in titrated doses as a preemptive analgesic. Morphine (5 mg) was administered if the monitor showed an NOL index between 25 and 45. Another 5 mg of morphine was given if the NOL index was above 45. The patient was awakened once the NOL index dropped under 25.

Before taking decisions about the management of hemodynamic disturbances, depth of anesthesia and adequacy of nociceptive therapy were confirmed using the BIS and NOL monitor. If the BIS and NOL index were in an acceptable range, like the control group, hypertension was treated with infusion of nitroglycerine and hypotension was treated with fluid bolus and IV ephedrine or IV infusion of phenylephrine wherever suitable.

For recovery room management in both groups, the verbal rating scale (VRS) was used for pain assessment during the participants’ stay in the recovery room (RR), and analgesic delivery was titrated to maintain the VRS below 3. IV morphine was used as an analgesic and delivered according to a standard local protocol. During the RR stay, the parameters were monitored as per recovery room discharge (RRD) score every 30 min. (Appendix A). Patients were discharged once the score was less than 4 for 30 min. PONV was treated according to the standard protocol.

### 2.12. Data Collection

The data were derived from routine monitors, i.e., BP, HR, ECG, and pulse oximetry, and documented on the anesthesia charts in real time. The charts were scanned and saved in the patient’s electronic journal. BIS and NOL indices saved in real time in the monitor memory were also documented on the anesthesia charts. All the data, including the detailed NOL index information from monitor memory, were further updated in Redcap software (last updated on march 2022). All the monitors were time-aligned before the induction of anesthesia. Various events occurring during anesthesia were annotated in the anesthesia charts, such as drug administration (including targets and doses) and surgical and anesthesia events (e.g., loss of consciousness, insertion of LMA, incision, end of surgery, eye opening, start and end of anesthesia).

### 2.13. Statistics

The power calculation is based on the observation that a nociception level cutoff value of 20 yeilds specificity and sensitivity values of 80% and 73%, respectively, for discriminating between nonpainful and painful stimuli. With the aim of a type I error < 0.05 and type II error < 0.20 and with the expectation of a greater than 30% reduction in opioid consumption, 28 patients were required in each group. To compensate for missing data and dropouts, we planned to include 30 patients in each group. Statistical calculations were performed using STATA (STATA BE/18, Stata Corp LLC, College Station, TX, USA) software. The normality of the data was checked using the D’Agostino–Pearson test for normally distributed data. The binomial data were analyzed using the chi^2^ test. The data were analyzed and are expressed as medians (interquartile ranges) for nonnormally distributed variables or as numbers and percentages for normally distributed data. The analysis of all obtained quality and hemodynamic data was performed offline after completion of the study. *p* < 0.05 was considered significant for all the statistical tests.

Further, to assist with certain aspects of the manuscript writing process, we used the large language model (LLM) ChatGPT (GPT-4o). However, the text was subsequently modified multiple times. Further, all analyses and interpretations were produced by the research team, and the content of the manuscript remains solely the responsibility of the authors.

## 3. Results

Patients were recruited from May 2022 to November 2023. Out of sixty patients initially enrolled, two were excluded (Figure 1). One patient from the study group was excluded as she needed endotracheal intubation, while one from the control group was excluded as she withdrew consent to participate in the study. Thus, fifty-eight patients (twenty-nine per group) were available for analysis. There was no statistically significant difference in height, weight, age, BMI, smoking, or alcohol consumption between the groups (Table 1).

Similarly, no difference was observed in the duration of surgery (74 min versus 78 min in the study group and control group, respectively; *p* = 0.257) or anesthesia time (117 versus 130 min in the study group and control group, respectively; *p* = 0.480). Additionally, there were no statistically significant differences in the length of hospital stay or maximum pain scores in the recovery room between the two groups (Table 2).

The study group required significantly less remifentanil (0.9 mg vs. 1.35 mg in the control group; *p* = 0.033) and morphine (2.5 mg vs. 5 mg; *p* = 0.013). Additionally, the median TCI remifentanil concentration was lower in the study group (5 ng/mL vs. 6 ng/mL in the control group; *p* = 0.001), which corresponds to a decreased opioid requirement when guided by NOL monitoring (Table 2). Propofol administration did not differ between the two groups (700 mg in study group, 670 mg in control group; *p* = 0.6515).

There were no significant differences between groups regarding PONV incidence. Eligibility for discharge from the recovery room did not differ significantly, with a median time of 60 min for both groups (*p* = 0.709). Other recovery parameters, such as maximum pain scores in the recovery room and overall recovery room stay duration, were also comparable between the groups.

At the three-month follow-up, PMPS symptoms were similar between the groups (17 patients in control group vs. 16 patients in study group; *p* = 0.792), indicating that the combination of PECS and parasternal blocks with NOL monitoring did not significantly affect long-term PMPS (Table 2).

None of the patients in either group received intravenous fluids, vasoconstrictors, inotropes, or atropine for hemodynamic disturbances. No patients in either group received perioperative blood transfusions or reoperations for bleeding.

The dynamic changes in these parameters in relation to each other in both groups are shown in the graphs in Figure 2a–d. The graphs demonstrate higher remifentanil infusion rates in the control group than in the study group. In the study group, there was an increase in the NOL index along with BIS and hemodynamic parameters during the last 30 min of the procedure, which, except for the BIS, decreased until the end of anesthesia. However, the NOL index remained stable in the control group until the end of the procedure.

Both groups had similar periods of excessive analgesia: the control group had excessive analgesia for 67% and the study group for 66% of the intraoperative period (*p* = 0.61). Adequate analgesia was also comparable between groups (27% vs. 25% of intraoperative period; *p* = 0.8). However, the study group experienced a longer duration of inadequate analgesia (10% vs. 7%; *p* = 0.008) (Table 3).

## 4. Discussion

This randomized controlled study demonstrated that NOL-guided analgesia reduced intraoperative opioid consumption without affecting outcomes such as pain on arrival, PONV, or PMPS symptoms. While the study group required less remifentanil and morphine, they experienced a statistically significant increase in the duration of inadequate analgesia compared to the control group.

The reduced opioid consumption in the study group can likely be attributed to the NOL monitor’s ability to detect and respond to nociceptive changes in real-time and the selection of the rate of TCI remifentanil, although, initiated as per the Minto protocol, the TCI rate was titrated under NOL guidance in the study group and using subjective clinical judgment in the control group [18]. Therefore, more opioid consumption in the control group may suggest a subjective tendency towards selection of higher TCI rates. However, the TCI rates in the control group never exceeded the recommended range and there were no noticeable hemodynamic disturbances, which may suggest too much remifentanil administration, were observed (Figure 2d). Therefore, questions arise regarding the recommended ranges of TCI infusion.

During TCI, drugs are delivered using a computer-controlled infusion pump to achieve a predictable concentration of the drug at the plasma level (TCIp) or at a specific site (brain) (TCIe) [19]. Although the TCI system offers the advantage of delivering precise doses to achieve the desired effects safely, the theoretical findings may not be easily applicable in clinical practice [19]. The technique underwent exponential development in the early 1990s, yet it has been underused for the last 20 years. The main challenge of the TCI is to define the target concentration and the plasmatic and site concentration ranges based on the level of effect we want to achieve. After 20 years of use of TCI systems, there is still a debate between various pharmacokinetic models, with significant differences among them regarding the time to peak effect, body mass integration in formulas, use of gender and age, and rare limitation of different commercial open TCI systems (high body mass index, pediatric models) [19]. The Minto model used for calculating the TCI of remifentanil infusion rate is based on the principle of achieving an effective site concentration sufficient to cause 50% of the maximum electroencephalogram (EEG) effect at steady state, where the calculation assumes a lean body mass of 55 kg [23]. As the ke0 (the rate constant for equilibrium between plasma and effect-site concentration) in the Minto model is derived from studies of EEG parameters as a measure of effect, it must be remembered that an EEG parameter does not necessarily equate with the onset of analgesic action. The increased plasma remifentanil concentrations required in the TCIe mode can be associated with a greater likelihood of chest wall rigidity and severe bradycardia via non-vagal mechanisms [24].

Additionally, the challenges in the TCI infusion rates may explain the higher opioid delivery and longer periods of excessive analgesia (i.e., 60%, Table 3) in the control group. However, similar periods of excessive analgesia in the study group, especially under NOL index guidance, need further investigation.

Further, statistically significantly lower intraoperative opioid use did not translate into differences in PONV, RR discharge eligibility, or PMPS symptoms, suggesting that NOL-guided opioid reductions are achievable, which may not necessarily impact clinical outcomes beyond the operating room.

## 5. Limitations

PONV has been widely studied as a multifactorial entity, and the reported incidence in female surgical populations is extremely variable among randomized clinical trials [12]. This implies that a wide patient population may be required for any study to confidently conclude effects on PONV. Therefore, the present study may appear underpowered.

With the observation of lower NOL indices (below 10) in the study group, especially in the presence of regional analgesia, TCI remifentanil may be expected to be reduced even lower. However, as an integral part of balanced anesthesia, remifentanil plays an important role in preventing potential airway complications, i.e., broncho- and laryngospasm [25]. Therefore, despite guidance on the depth of anesthesia through the BIS, the possibility of not reducing the TCI remifentanil rate due to apprehension of potential airway complications cannot be denied [26,27,28].

According to the monitor guidelines, supplemental analgesics are suggested when the NOL is >25 for at least 2 min (especially after administration of sympathomimetic drugs), as only after 2 min does the index suggest a clinically relevant nociceptive reaction. The 2-minute delay increases the NOL index response time of the administered analgesic. This may result in longer cumulative periods of inadequate analgesia. However, no hemodynamic repercussions or need of higher doses of analgesics were observed in the study group.

## 6. Conclusions

NOL monitor-guided analgesic delivery reduces intraoperative opioid consumption. No difference was demonstrated on PONV, eligibility to discharge from the recovery room, or PMPS symptoms. Future large RCTs based on NOL monitors are required to investigate the observation of longer periods of intraoperative excessive analgesia and advantages of intraoperative opioid optimization, especially in the context of PONV and long-term complications in breast cancer surgery.

## Figures and Tables

**Figure 1 medicina-60-01921-f001:**
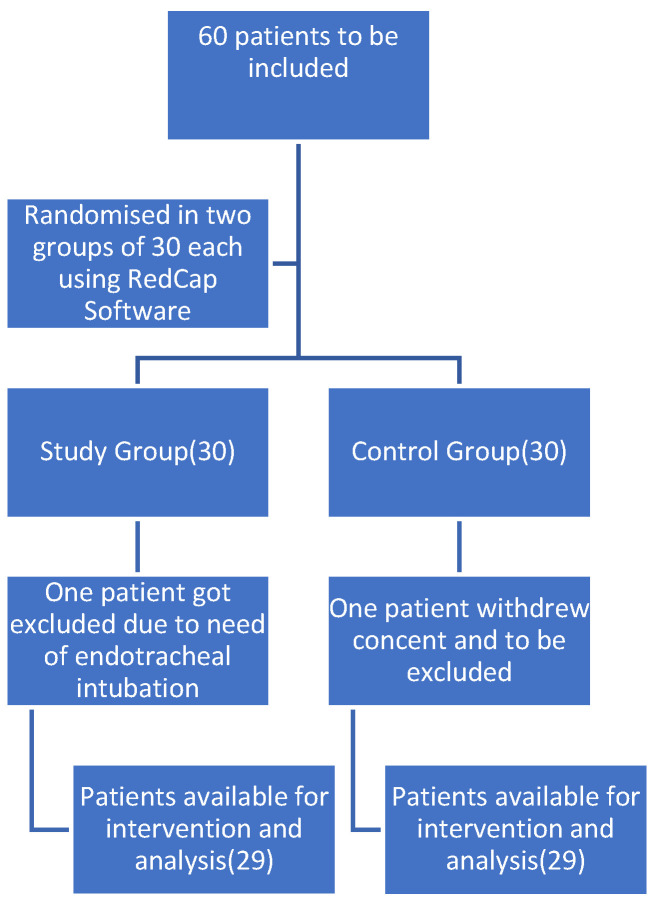
CONSORT flow diagram.

**Figure 2 medicina-60-01921-f002:**
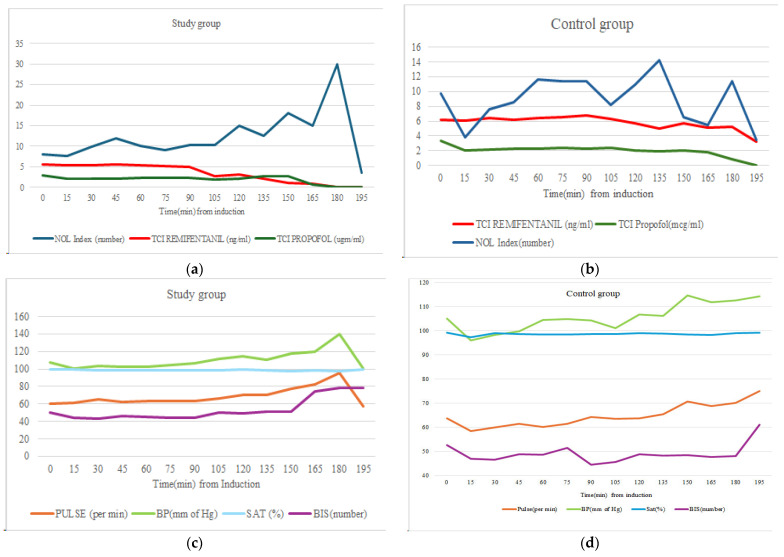
Dynamic interplay between TCI remifentanil, TCI propofol, NOL index, BIS, blood pressure (BP), saturation (SAT), and pulse in study and control groups during the intraoperative period. (**a**,**c**) Dynamics in study group; (**b**,**d**) corresponding changes in control group. NOL index and TCI remifentanil were stable throughout the procedure compared to control group. The graphs demonstrate higher remifentanil infusion rates in the control group than in the study group. No major hemodynamic irregularities were observed in either group.

**Table 1 medicina-60-01921-t001:** Demographics.

Parameters	Control (*n* = 29)	Study (*n* = 29)	*p* *
Height (cm)	160 (165–168)	162 (166.5–170)	0.0822
Weight (kg)	61 (65–73)	59.5 (71–85)	0.2403
ASA 2–4 *n* (%)	3 (10.34)	3 (10.34)	0.5172
Age (years)	53 (71–75)	52 (64–75)	0.6431
BMI kg/m^2^	21.95 (24.65–28.55)	22.3 (26.25–31.2)	0.5174
Smoking *n* (%)	7 (24.13)	8 (27.58)	0.243 ^#^
Alcohol *n* (%)	12 (41.38)	10 (34.48)	0.661

All values are presented as medians with IQRs. BMI = body mass index, ^#^ Fisher’s chi2 exact. ASA = American Society of Anesthesiology. * Wilcoxon rank-sum (Mann–Whitney) test.

**Table 2 medicina-60-01921-t002:** Observations.

Parameters	Control	Study	*p* *
Total surgery time (min)	78 (102–122)	74 (90–111)	0.2577
Total anesthesia time (min)	130 (149–180)	117 (147–170)	0.4805
Total propofol (mg)	670 (830–1070)	700 (902–1090)	0.6515
Total remifentanil (mg)	1.35 (2.1–3)	0.9 (1.38–1.8)	0.033
TCI remifentanil (ng/mL)	6 (5–7)	5 (4–6)	0.0001
Total intraoperative morphine (mg)	5 (5–10)	2.5 (0–5)	0.0013
VRS on arrival in RR (n)	0 (0–4)	3 (0–5)	0.2015
Postoperative morphine (mg)	3.75 (0–7.5)	5 (0–7.5)	0.8277
RR stay duration (min)	95 (75–120)	87.5 (72–112.5)	0.4871
Eligibility to discharge (min)	60 (60–90)	60 (30–60)	0.7090
Length of hospital stay (hours)	10.4 (9–24)	11.2 (8.5–23)	0.656
PMPS symptoms *n* (total)	17 (27)	16 (28)	0.7927 ^#^
Nausea on arrival in RR *n* (total)	0 (29)	1 (29)	0.2994
PONV during stay in RR *n* (total)	2 (29)	3 (29)	0.5536

TCI = target-controlled infusion, VRS = verbal rating scale. RR = recovery room. PMPS = postmastectomy pain syndrome. PONV = postoperative nausea and vomiting. All values are presented as medians with IQRs. * Wilcoxon rank-sum (Mann–Whitney) test. ^#^ Two-sample Wilcoxon rank-sum (Mann–Whitney) test.

**Table 3 medicina-60-01921-t003:** Intraoperative analgesia analysis.

State of Analgesia (NOL Index Value)	Control (% of Intraoperative Time)	Study (% of Intraoperative Time)	*p* *
Adequate analgesia (10–27)	27 (18–35)	25 (17–34)	0.61
Excessive analgesia (<10)	67 (57–77)	66 (59–75)	0.86
Inadequate analgesia (>27)	7 (5–8)	10 (7–12)	0.008
Inappropriate analgesia (>27 + <10)	74 (65–82)	74 (66–83)	0.8092

The state of analgesia as a percentage of the intraoperative time at which the NOL index remained above 27, below 10, and between 10 and 27. * Two-sample Wilcoxon rank-sum (Mann–Whitney) test. All values in medians and IQRs.

## Data Availability

The data that support the findings of this study are available from the authors, but restrictions may apply to the availability of these data. The data are, however, available from the authors upon reasonable request.

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
