# Peer review of "Influence of Nociception Level Monitor (NOL)-Guided Analgesic Delivery on Perioperative Course in Breast Surgeries: A Randomized Controlled Trial"

_medicina, 2024, doi:10.3390/medicina60121921_

Round 1

Reviewer 1 Report

Comments and Suggestions for Authors

This study explores the influence of nociception level monitor (NOL) guided analgesic delivery on the perioperative course in breast surgeries. The findings present promising insights into the potential benefits of utilizing NOL for optimizing analgesic management and enhancing patient outcomes. However, there are some areas that require attention and improvement to strengthen the overall quality of the research.

1. 'Various strategies, including ultrasoundguided pectoral nerve blocks (PECS II) and parasternal blocks, have been explored to minimize opioid use.[13, 14]'' The statement notes that various strategies, including ultrasound-guided pectoral nerve blocks (PECS II) and parasternal blocks, have been explored to minimize opioid use. However, both references pertain specifically to pectoral blocks, and there appears to be a limited number of studies related to parasternal blocks in breast surgery. To enhance clarity, you can either remove "parasternal blocks" from the sentence or include a suitable reference that supports its relevance in this context.

2. Please start the "Materials and Methods" section with the 'Study Design' subheading and adjust its placement accordingly. This will enhance the clarity and organization of the section.

3. The terms "PECS II and parasternal nerve blocks" are frequently used throughout the text. However, it is important to note that there are actually three blocks involved. To adhere to current nomenclature standards, these should be revised to "interpectoral, pectoserratus, and superficial pecto-intercostal blocks," as published in the literature on regional anesthesia and pain management (RAPM). Please ensure that the new nomenclature is used consistently throughout the manuscript.

4. The sentence "In both groups, PECS II and parasternal blocks were used to enhance analgesia as part of a multimodal approach..." implies that the blocks were performed prior to surgery. To improve clarity, we suggest adding a subheading titled "Block Performance." In this section, please detail when the blocks were performed (e.g., before or after intubation), the sedation status (e.g., under sedation or general anesthesia), the ultrasound technique used (e.g., transducer type), the needle used, the volume and concentration of the medication administered, and any specific drugs utilized. This additional information will enhance the understanding of the analgesic techniques employed in the study.

5. The description of the PECS II block contains inaccuracies. Specifically, it is not possible to block the long thoracic nerve between the pectoralis major and minor due to anatomical constraints. Additionally, the information presented is misleading in the context of regional anesthesia. (many mistakes)

Furthermore, the reference to "Naropine" is inappropriate, as it should not be listed by its trade name; instead, please specify the active ingredient and concentration.

For clarity and accuracy, I recommend creating a new section to detail the following for each block:

  • The volume administered (in mL) for both the PECS II blocks and the parasternal block.
  • The correct anatomical targets for each block.
  • The specific ultrasound technique and needle used.

This will ensure the information presented is both accurate and informative.

6. There is a formatting issue at the beginning of page 6, leading to misalignment of the lines. Please correct the alignment to ensure consistency and readability throughout the manuscript.

7. The conclusion could be improved by discussing the clinical significance of reducing intraoperative opioid consumption, the broader implications for multimodal analgesia, and recommendations for practice regarding NOL monitoring. Additionally, it could propose future research directions, such as examining the long-term effects of reduced opioid use on patient outcomes.

8. There is an inconsistency in the classification of ASA scores within the manuscript. The article references ASA I-III, while the clinical trial mentions ASA I-IV v (ClinicalTrials.gov ID: NCT05546021. This discrepancy should be corrected to ensure consistency throughout the text. Please clarify whether the study includes ASA I-III or ASA I-IV and make the necessary adjustments in both sections.

9. The manuscript does not specify the time frame during which the study was conducted. The clinical trial registration (NCT) indicates that the study began in September 2023, but its status remains "Enrolling by invitation." It is crucial to update the NCT status to "Completed" and include the start and end dates of the study in the manuscript.

Author Response

Comment 1. 'Various strategies, including ultrasound guided pectoral nerve blocks (PECS II) and parasternal blocks, have been explored to minimize opioid use.[13, 14]'' The statement notes that various strategies, including ultrasound-guided pectoral nerve blocks (PECS II) and parasternal blocks, have been explored to minimize opioid use. However, both references pertain specifically to pectoral blocks, and there appears to be a limited number of studies related to parasternal blocks in breast surgery. To enhance clarity, you can either remove "parasternal blocks" from the sentence or include a suitable reference that supports its relevance in this context.

Response: Thank you for your observation. We agree that the reference no. 13 and 14 both represent PECS II block. We have replaced reference no. 14 with another reference which supports the Use of superficial pecto intercostal block. ( Page2. Line 12. Reference nr. 14 replaced by new. The reference list is updated)

Comment 2. Please start the "Materials and Methods" section with the 'Study Design' subheading and adjust its placement accordingly. This will enhance the clarity and organization of the section.

Response: Thank you for the suggestion. We have made advised changes. (Page 2. Line 24,25,26,27 )

Comment 3. The terms "PECS II and parasternal nerve blocks" are frequently used throughout the text. However, it is important to note that there are actually three blocks involved. To adhere to current nomenclature standards, these should be revised to "interpectoral, pectoserratus, and superficial pecto-intercostal blocks," as published in the literature on regional anesthesia and pain management (RAPM). Please ensure that the new nomenclature is used consistently throughout the manuscript.

Response: Thank you for your reflection. We agree with your observation. We have made suggested changes. As we observe that, this nomenclature is not used routinely in every day practice yet, we chose to use the old name i.e. PECSII block in a parenthesis in its first appearance in the abstract and main manuscript. (Page1 line 29,30)

Comment 4. The sentence "In both groups, PECS II and parasternal blocks were used to enhance analgesia as part of a multimodal approach..." implies that the blocks were performed prior to surgery. To improve clarity, we suggest adding a subheading titled "Block Performance." In this section, please detail when the blocks were performed (e.g., before or after intubation), the sedation status (e.g., under sedation or general anesthesia), the ultrasound technique used (e.g., transducer type), the needle used, the volume and concentration of the medication administered, and any specific drugs utilized. This additional information will enhance the understanding of the analgesic techniques employed in the study.

Response: Thank you for your suggestion. We have made a separate subheading with the name” Block performance”. We have described the block administration in adequate details. (Page 4 line 1-16)

Comment 5a. The description of the PECS II block contains inaccuracies. Specifically, it is not possible to block the long thoracic nerve between the pectoralis major and minor due to anatomical constraints. Additionally, the information presented is misleading in the context of regional anesthesia. (many mistakes)

Response:  Thank you for the observation. We have rewritten the paragraph under “ Block Performance “ section.  Here we wish to comment that, Bianco.et.al, (2012) have described the block in detail. Here, Bianco states that, the block is placed in the intention to block all nerves between the three muscles including Long thoracic nerve. We agree with the reviewer that, it is difficult to block the nerve due to its anatomical position. The long thoracic nerve is not typically directly targeted during a PECS II block, but it may be incidentally affected due to its anatomical location near the serratus anterior muscle. In standard practice, the long thoracic nerve lies posterior to the injection plane between the pectoralis minor and serratus anterior muscles i.e. on its superficial surface and runs on the lateral aspect of the thoracic cage. Thus, we presume that it is not impossible.

We have modified the paragraph as per the reviewer´s suggestions to avoid confusion.

Comment 5b. Furthermore, the reference to "Naropine" is inappropriate, as it should not be listed by its trade name; instead, please specify the active ingredient and concentration.

Response: Thank you for the observation. We have made the correction.

Comment 5c.For clarity and accuracy, I recommend creating a new section to detail the following for each block:

The volume administered (in mL) for both the PECS II blocks and the parasternal block.

The correct anatomical targets for each block.

The specific ultrasound technique and needle used.

This will ensure the information presented is both accurate and informative.

Response: Thank you for the suggestion. We have written a separate paragraph on block performance. The paragraph addresses all the suggestions. ( Page 4 line 1-16)

Comment 6.There is a formatting issue at the beginning of page 6, leading to misalignment of the lines. Please correct the alignment to ensure consistency and readability throughout the manuscript.

Response: The manuscript has been transferred from Word document in to the journal´s template. This has given multiple typos and formatting issues. We have tried to address them. (Page 6 Line 41- 55)

Comment 7. The conclusion could be improved by discussing the clinical significance of reducing intraoperative opioid consumption, the broader implications for multimodal analgesia, and recommendations for practice regarding NOL monitoring. Additionally, it could propose future research directions, such as examining the long-term effects of reduced opioid use on patient outcomes.

Response: Thank you for the suggestion about discussion of clinical significance of reducing intraoperative opioid consumption in conclusion. Here, we would be like to point out that, in our study, we have not observed any clinically significant advantage of the reduced intra operative opioid consumption in immediate as well as extended postoperative period. We will certainly mention proposal of future clinical research. ( Page 9. Line 36-38)

Comment 8. There is an inconsistency in the classification of ASA scores within the manuscript. The article references ASA I-III, while the clinical trial mentions ASA I-IV v (ClinicalTrials.gov ID: NCT05546021. This discrepancy should be corrected to ensure consistency throughout the text. Please clarify whether the study includes ASA I-III or ASA I-IV and make the necessary adjustments in both sections.

Response: Thank you for pointing out the discrepancy. We have not included any patient with ASA 4 grading.  We have updated the manuscript accordingly. ( Page2. Line 38. & Page5,Table 1 .)

Comment 9. The manuscript does not specify the time frame during which the study was conducted. The clinical trial registration (NCT) indicates that the study began in September 2023, but its status remains "Enrolling by invitation." It is crucial to update the NCT status to "Completed" and include the start and end dates of the study in the manuscript.

Response: Thank you again for pointing out the discrepancy. We have updated the file in the clinicaltrails.gov

Reviewer 2 Report

Comments and Suggestions for Authors

General comments

The authors compared NOL guided remifentanil administration with standard remifentanil administration in breast surgeries. Although, the study is interesting, I have several major concerns. First, please follow the CONSORT guidelines (PMID: 20335313). Second, what is the clinical meaning of the results? In other words, why the study is important? Third, please clearly describe the “standard protocol”.

Major concerns

#1. Please follow the CONSORT guidelines (PMID: 20335313).

#2. The NOL guided remifentanil administration reduced total remifentanil dose. The NOL guided administration resulted poor analgesic control (Table 3). What is the clinical meaning of the results? In other words, why the study is important?

#3. The authors use the words “standard practice” (Page 2, Line 32), “standard protocol” (Page 3, Lines 25, 38, 46), “framework departmental guidelines” (Page 3, Line 48), and “standard local protocol” (Page 4, Line 10). Please describe the “standard protocol” in detail, so that other researchers can replicate the study.

Minor comments

#1. Please use generic name of Naropine (Page 3, Line 43).

#2. Please remove the values already shown in Tables (Page 4, Lines 48).

#3. Why did the authors select morphine doses (Page 3, Line 49, Page 4, Line 4)? Please include the reasons.

#4. What is “inj.” (Page 3, Line 49)?

#5. Error bars are missing in Figures 1-4.

Author Response

 Major concerns

Comment 1. Please follow the CONSORT guidelines (PMID: 20335313).

 Response: Thank you for the suggestion. We have updated the manuscript throughout to compile with suggestions. The consort diagram Is now followed and all headings updated accordingly. (Page3, Line1-4)

Comment 2. The NOL guided remifentanil administration reduced total remifentanil dose. The NOL guided administration resulted poor analgesic control (Table 3). What is the clinical meaning of the results? In other words, why the study is important?

 Response: We understand the confusion. We analyzed our current intraoperative analgesic delivery practice with an objective nociception monitor. The decision and amount of analgesics was controlled by the NOL monitor only until the conventional hemodynamic parameters were in acceptable range. The expression “poor analgesic control “originates from manufacturer’s classification of the analgesia which is based on the range of NOL index.  We have in our daily clinical practice noticed NOL index higher than 45 without any hemodynamic disturbance i.e. clinical suspicion. The observation of  “poor analgesic control”  translate NOL index over 25 or under 10 and not necessary clinically ( conventionally) assessed pain. As per table 3, despite objective monitoring of nociception, study group showed similar duration of inappropriate analgesia. This reflects that, further changes in practice are warranted. 

Comment 3. The authors use the words “standard practice” (Page 2, Line 32), “standard protocol” (Page 3, Lines 25, 38, 46), “framework departmental guidelines” (Page 3, Line 48), and “standard local protocol” (Page 4, Line 10). Please describe the “standard protocol” in detail, so that other researchers can replicate the study.

Response: Thank you for your observation. We apologize for the confusion created by subheading. We have changed the subheading “General treatment in both groups “ to “ Standard anesthesia regimen for both the group”( Page3,Line 27. Page3, line 47. Page3,line 49,50. Page 4,line 17,18.)

Minor comments

Comment 1. Please use generic name of Naropine (Page 3, Line 43).

Response: Thank you for your observation, We have corrected this in the manuscript.

Comment 2. Please remove the values already shown in Tables (Page 4, Lines 48).

Response: As the manuscript has been modified in to the journals template, we have now updated it correctly.(Page 5line 26 -31)

Comment 3. Why did the authors select morphine doses (Page 3, Line 49, Page 4, Line 4)? Please include the reasons.

Response: Thank you for your observation. We have added explanation in the respected paragraphs i.e subheading “Control group” and “Standard group.”( Page 4 line 20- 27 and line 31 – 41)

Comment 4. What is “inj.” (Page 3, Line 49)?

Response: We apologize for the observation. But we could not find the mentioned “inj” at the described position. It may be due to transfer of the manuscript from original word document to the journal´s template. We have updated the manuscript accordingly.

Comment 5. Error bars are missing in Figures 1-4.

Response: Thank you for your observation. Our intention for the Fig 1-4 was just to portray the interplay between the anesthetic drugs and conventional hemodynamic parameters in both the groups. As per our judgment, further description would not add to the required information and would in fact make the presentation further complicated. No changes were made.

Round 2

Reviewer 1 Report

Comments and Suggestions for Authors

It is acceptable in its current form. However, it should be checked for spelling before printing.

Author Response

Comment 1: It is acceptable in its current form. However, it should be checked for spelling before printing.

Response: Thank you for the encouraging comments. We have re-screened the manuscript using grammar and spellcheck softwares. Further, our english writing expert have also gone-through the manuscript. We have made some changes for better comprehension.

The changes can be seen on the following pages and lines.

( Page 1- line 30;  Page 3 - line 2,38,40,44,45-50;Page 4-line 13,14,15,27-32,34, 41,45. Page5-line 3,32,; Page6-line 1, 11,12,13, 26.Page 8- 23,24,25,26,27,28 Page9-line 2,4,5)  

Reference nr.16 and 22.

Reviewer 2 Report

Comments and Suggestions for Authors

General comments

The authors have improved the manuscript little. Also, the authors did not answer my major concerns. Please read and follow the CONSORT guidelines carefully. There is no flow diagram, no CONSORT checklist. Again, what is the clinical meaning of the results? In other words, why the study is important?

Major concerns

#1. Please follow the CONSORT guidelines (PMID: 20335313).

#2. What is the clinical meaning of the results? In other words, why the study is important?

Minor comments

#1. Again, why did the authors select morphine doses? I do not see the REASONS. Please include the reasons.

#2. What is “inj.” (Page 4, Line 32)?

#3. I do not understand Figures 1-4. There are no error bars, no legends.

Comments on the Quality of English Language

There are quite some odd wordings and use of grammar. I do suggest to have some external language editing done by a person familiar with the field.

Author Response

Thank you for the comments. We appreciate your concerns.

We have stated the responses  to all the concerns as follows

Reviewers primary comments

The authors have improved the manuscript little. Also, the authors did not answer my major concerns. Please read and follow the CONSORT guidelines carefully. There is no flow diagram, no CONSORT checklist. Again, what is the clinical meaning of the results? In other words, why the study is important? 

Major concerns

#1. Please follow the CONSORT guidelines (PMID: 20335313).

Response: We have added consort flow chart as Fig.1. (page 6, line 1). We have mentioned the use of consort guidelines in manuscript and substantiated it with the reference mentioned by you.                   ( ref.no.16). ( Page 2-line33-35)
We have added the completed consort checklist as supplementary Material 1. We have made sure that we have included all the important elements of the consort guidelines  in the manuscript.

#2. What is the clinical meaning of the results? In other words, why the study is important?

Response: We apologise for not being able to convey our clear message in previous response.

The results show that, NOL guided analgesic delivery reduce intraoperative opioid consumption. This was our primary outcome measure. We further investigated if this reduced opioid consumption reflects in early discharge from the recovery room, reduced postoperative pain, reduced PONV and PMPS. Our study revealed that, these postoperative parameters of interest remain unchanged despite reduced opioid consumption. So, technically, there is no advantage of the use of NOL monitor except some savings in cost of opioids.

The importance of this study: Due to relative simplicity of the mastectomy procedure, there were requests from the breast surgery department to assess if the patients could be discharged from the hospital on the same day, which presently remain in the hospital until next day primarily due to PONV and postoperative pain. Therefore, the study was designed to assess if use of NOL monitor offer both logistic and economical advantage to the institute (less opioids, shorter recovery occupancy etc.) as well as to patients ( less PONV, early discharge, no PMPS).

Minor concerns

#1. Again, why did the authors select morphine doses? I do not see the REASONS. Please include the reasons.

Response: We apologise again for not being able to convey clear explanation for the use of morphine. We had mentioned in the previous version of the manuscript that; Morphine has been used as preemptive analgesic in order to avoid pain in immediate postoperative period. (Page 4, line 27-32) It is our observation that, despite PECSII block, the surgical work deep in axillary region may give severe pain, which is masked during surgery by remifentanil but may surface when the patient is awake. This is because, PECS II do not block all the nerves in the surgical area. This has been beautifully demonstrated by  Kim DH et.al** ( reference nr 22 in manuscript).  Therefore, use of preemptive analgesics (and in our institute, Morphine) in breast surgical patients is a standard practice in all Danish hospitals.  We have added the explanation in the manuscript.(Page4 line 27-32). And also substantiated the statement with the mentioned reference.( Ref. nr.22)

Here we are little confused about the question. Is it “why we chose Morphine?” or “Why we chose the amount of morphine which we decided to give?”

We have already answered the first question. We will try to explain about the dose. Use of inj. Morphine 10 mg is a standard practice in these type of surgeries, therefore  we used it in control group. In study group, we chose to first confirm if the patient has pain through NOL index and then supplement Morphine accordingly in graded doses. This was to reveal the advantage of objective nociception monitoring.

#2. What is “inj.” (Page 4, Line 32)?

Response: We apologise for the typo. We have corrected it in the manuscript. It is Inj. Morphine.(Page 4-line41)

#3. I do not understand Figures 1-4. There are no error bars, no legends. 

Response: We have modified the numbers of the figures and have added a legend as per the advice. The manuscript also have used the figures to support some statements. ( page 7 , line nr. 33-39 & page 9, line 38) Regarding the error bars, we have reassessed its need against its disadvantages. We still believe that, any further addition to the figures will make the comprehension difficult. The objective of the figures is to display the interplay between various parameters and show the difference in the two groups. No statistical calculation or assessment has been implied in the figures and no inference has been derived.

Comments on the Quality of English Language

There are quite some odd wordings and use of grammar. I do suggest to have some external language editing done by a person familiar with the field.

Response: We appreciate the concern. We have gone through the manuscript again. And made some changes where the comprehension may find difficult. The manuscript has been cross checked by the english writing expert of the university .

Further, for better comprehension we have added the chart of RRD score as supplementary material 2.

The changes made in language for better comprehension can be seen on the following pages and lines.

( Page 1- line 30;  Page 3 - line 2,38,40,44,45-50;Page 4-line 13,14,15,27-32,34, 41,45. Page5-line 3,32,; Page6-line 1, 11,12,13, 26.Page 8- 23,24,25,26,27,28 Page9-line 2,4,5)  

(Reference nr.16 and 22)